# Nosocomial Outbreak of *Ralstonia pickettii* Infections Likely Linked to Saline Solutions in Germany from August 2023 to March 2024—Challenges in Medical Product-Related Outbreaks

**DOI:** 10.3390/microorganisms13092102

**Published:** 2025-09-09

**Authors:** Mirco Sandfort, Anja von Laer, Stefanie Kampmeier, Inga Eichhorn, Silver A. Wolf, Maximilian Driller, Vladimir Bajić, Somayyeh Sedaghatjoo, Stephan Fuchs, Steffen Engelhart, Nico T. Mutters, Esther P. Cónsul-Tejero, Josef Zündorf, Tim Eckmanns, Michael Hogardt, Sebastian Haller

**Affiliations:** 1Infectious Disease Epidemiology, Robert Koch Institute, 13353 Berlin, Germany; laera@rki.de (A.v.L.); eckmannst@rki.de (T.E.); hallers@rki.de (S.H.); 2Infection Control and Antimicrobial Stewardship Unit, University Hospital Würzburg, 97080 Würzburg, Germany; kampmeier_s@ukw.de; 3Institute for Hygiene and Microbiology, University of Würzburg, 97070 Würzburg, Germany; 4Genome Competence Centre (MF 1), Robert Koch Institute, 13353 Berlin, Germany; eichhorni@rki.de (I.E.); wolfs@rki.de (S.A.W.); drillerm@rki.de (M.D.); bajicv@rki.de (V.B.); sedaghatjoos@rki.de (S.S.); fuchss@rki.de (S.F.); 5Institute for Hygiene and Public Health, University Hospital Bonn, 53127 Bonn, Germany; steffen.engelhart@ukbonn.de (S.E.); nico.mutters@ukbonn.de (N.T.M.); 6Federal Institute for Drugs and Medical Devices (BfArM), 53175 Bonn, Germany; esther-patricia.consul-tejero@bfarm.de (E.P.C.-T.); josef.zuendorf@bfarm.de (J.Z.); 7German National Consiliary Laboratory on Cystic Fibrosis Bacteriology, 60596 Frankfurt am Main, Germany; michael.hogardt@unimedizin-ffm.de; 8Institute of Medical Microbiology and Infection Control, University Hospital Frankfurt, Goethe University, 60596 Frankfurt am Main, Germany

**Keywords:** saline solution, *Ralstonia pickettii*, blood culture, bacteremia, sepsis, disease outbreaks, multilocus sequence typing

## Abstract

In December 2023, the World Health Organization reported an Australian nosocomial *Ralstonia pickettii* outbreak caused by contaminated saline solutions, causing bloodstream infections. The Antibiotic Resistance Surveillance, a Germany-wide laboratory network, has identified five *R. pickettii*-bacteraemia cases since August 2023 (0–1 annually 2019–2022), prompting an outbreak investigation to identify the source. We defined a case as a person with *R. pickettii* in any material (possible) or in blood culture (probable), confirmed if in the genetic outbreak cluster. We implemented nationwide *R. pickettii* surveillance, subjected isolates to core genome multilocus sequence typing, and compared cases by exposure, including medical products. From August 2023 to June 2024, we detected 25 cases. For eighteen cases, isolates were available, and sequences from six cases clustered within 6 allelic differences but showed >43 allelic differences from Australian outbreak-associated sequences. These confirmed cases occurred in three hospitals across three federal states between October 2023 and March 2024, linked only by saline solution exposure. Four products and 11 lots matched across two of three hospitals. Retain sample testing remained negative. The *R. pickettii* outbreak in Germany was likely linked to saline solutions, although no product/lot was confirmed. Tracing products/lots was particularly challenging. Patient-level documentation of medical products/lots and official mandates for the testing of retained samples could improve product traceback in future outbreaks.

## 1. Introduction

*Ralstonia pickettii* is a species of non-fermentative, Gram-negative bacteria that is commonly found in environmental sources such as soil and water [1]. Although rarely isolated from clinical specimens, *Ralstonia* spp. can cause occasional severe infections, including bloodstream infections or meningitis [2]. Antimicrobial resistance also occurs frequently, further challenging treatment options [3]. Immunocompromised individuals are thereby at particular risk for severe infection [4,5,6].

*R. pickettii* can persist and grow within water and saline solutions, form biofilms, and pass through 0.2 µm filtration systems commonly used in product sterilization processes [7,8,9]. This species is thus prone to contaminating water sources, industrial productions, and subsequently solution-based medicines and medical products [2]. Consequently, *R. pickettii* is an emerging opportunistic pathogen in healthcare-associated infections, with several reported nosocomial outbreaks [2,10]. Many of the reported cases involve severe infections, including bloodstream infections [11,12,13,14,15,16,17,18,19,20], and multiple outbreaks have been reported to be caused by contaminated saline solutions [12,13,16].

In December 2023, a nosocomial outbreak of *R. pickettii* was reported by Australian authorities to the World Health Organization and posted on the Event Information Site, a public health alert platform for member states. Across Australia, 55 *R. pickettii* cases had been detected between August and December 2023, and 84% of cases were exposed to sodium chloride 0.9% solution [21]. Isolates from cases and the product were genetically identical [21,22].

In Germany, *R. pickettii* is not mandatorily notifiable. Thus, after the alert, we examined the laboratory-based Antibiotic Resistance Surveillance (ARS, https://amr.rki.de (accessed on 19 December 2023)) system for a potential increase in new detections. In December 2023, we found five *R. pickettii*-positive blood cultures from hospitals across two federal states in Germany since August 2023, in contrast to at most one detection annually throughout 2017–2022 and no increase in other materials. This prompted an immediate outbreak investigation on a national level. The aim was to identify the source and prevent further cases by (i) confirming the suspected outbreak; (ii) identifying additional cases and common exposures; and (iii) investigating potential connections between the outbreaks in Germany and Australia.

## 2. Material and Methods

### 2.1. Outbreak Case Definition

A person with a positive specimen for *R. pickettii* in any material in Germany from August 2023 to June 2024 was initially defined as a possible outbreak case and subsequently categorized as follows:

A probable outbreak case if an isolate was identified in blood culture but was not available for sequencing; 

A confirmed outbreak case if an isolate was available for sequencing and could be linked to the genetic outbreak cluster within an allele distance of 6.

If the sequenced isolate was not associated with the German outbreak cluster, the case was excluded from further investigation.

### 2.2. Case Findings and R. pickettii Surveillance

Following the signal detection in December 2023, we implemented nationwide surveillance considering all detections of *R. pickettii* since August 2023 notifiable as part of a nosocomial outbreak according to the Protection against Infection Act (IfSG). Consequently, clinical laboratories needed to report to the public health authority any detection of *R. pickettii* in a patient sample as a result of routine, standardized microbiological diagnostics, e.g., by mass spectrometry or automated biochemical tests. All submitted isolates were verified for species identification by using matrix-assisted laser desorption/ionization time-of-flight (MALDI-TOF) (VITEK-MS Prime, bioMérieux, Marcy l’Etoile, France) based on knowledge database V3.2 (covering *Ralstonia insidiosa*, *Ralstonia mannitolilytica*, and *R. pickettii*). We informed major stakeholders in public health, microbiology, and hospital hygiene throughout January 2024 to notify respective cases and to send isolates for molecular investigation [23,24,25].

Patients with *R. pickettii* in blood culture from August to December 2023 were retrospectively included. For sequence-based confirmation, isolates were requested. Cases were retrospectively identified through ARS, a voluntary laboratory network in Germany that encompasses the detection of microorganisms across all specimens of routine diagnostic microbiology. In 2023, ARS covered 47% of hospitals and 35% of outpatient clinics. While only laboratories with continuous participation in ARS throughout 2017–2023 were considered to detect the outbreak signal, all laboratories were considered for subsequent case finding.

We informed public health authorities within the European Union, the United Kingdom, and Australia about the outbreak and shared sequencing data for comparison.

For all cases, we collected information including gender, age, comorbidities, especially immunosuppression, symptoms, focus of infection, clinical indications for diagnostics, and date of sampling. For confirmed cases, we collected and compared prior exposures: medicines, procedures, indwelling devices, and medical products. Data on specific products and product lots for common exposures were gathered on the hospital, ward, or patient level depending on data availability and compared between facilities.

### 2.3. Whole-Genome Sequencing

Genomic DNA of cultured bacteria was extracted using the DNeasy UlraClean 96 kit (Qiagen, Venlo, The Netherlands). Library preparation was performed using the Illumina Nextera XT DNA Library Preparation-kit (Illumina Inc., San Diego, CA, USA), optimized for small genomes. Half the volume specified by the manufacturer, i.e., a total of 1 ng of genomic DNA (gDNA), was utilized. The libraries were normalized based on quantification using concentration measured with the Qubit 1X dsDNA High Sensitivity Assay-kit (Thermo Fisher Scientific Inc., Wilmington, DE, USA) using a Qubit Flex Fluorometer (Invitrogen by Thermo Fisher Scientific, Wilmington, DE, USA) and the average fragment size using the TapeStation 4150 (Agilent Technologies, Inc., Waldbronn, Germany) with the High Sensitivity D1000 ScreenTape-assay. The normalized libraries were pooled according to the manufacturer’s protocol and sequenced on an Illumina iSeq 100 using i1 Reagent v2 with a 2 × 150 bp read configuration.

Six samples (13_B3672, 23_B33984, BK115556, BB_XA27_0046, BB_XB07_0060, and OR2283) were additionally sequenced using the Oxford Nanopore platform (Oxford Nanopore Technology Ltd., Oxford, UK). Library preparation was performed with the Ligation Sequencing V14-kit (SQK-LSK114; Oxford Nanopore Technologies, Oxford, UK), combined with native barcoding (EXP-NBD196; Oxford Nanopore Technologies, Oxford, UK), following the manufacturer’s protocol. The libraries were loaded onto R10.4.1 flow cells and sequenced on GridION Mk1 for 72 h. Basecalling was performed using MinKNOW v23.07.12, enabling super-accurate basecalling, barcoding on both ends, and mid-read barcode filtering.

### 2.4. Genomic Reconstruction

Short-read sequences were quality-controlled, removing adapter sequences and assessing read statistics via fastp v0.23.4 [26], and de novo assembled using SPAdes (v3.15.5) [27] within the GARI pipeline (v1.0, https://github.com/rki-mf1/GARI (accessed on 8 February 2024)).

Long-read sequences were subsampled to a coverage of ~200x using Filtlong (v0.2.1) (https://github.com/rrwick/Filtlong (accessed on 24 February 2024)), including only long, high-quality reads. De novo assemblies were subsequently generated with Flye (v2.9.3) [28] and polished in conjunction with short-reads through Polypolish (v0.6.0) [29]. For samples with insufficient long-read coverage to generate high-quality assemblies, Unicycler (v0.5.0) [30] was used to produce short-read-first hybrid assemblies.

Eight publicly available *R. pickettii* sequences were included for genomic comparison. These also included (i) an Australian outbreak sample (NCBI BioProject PRJNA1040313) and (ii) samples from the UK investigation related to the Australian outbreak [31]. The overall 28 samples included in this study are listed in Appendix A.

### 2.5. Bioinformatic Analysis

Ad hoc core genome multilocus sequence typing (cgMLST) scheme generation and allele calling were performed with chewBBACA (v3.3.10, [32]) based on 28 samples and a prodigal training file derived from the NCBI reference genome for *R. pickettii* (GenBank: GCA_902374465.1; Assembly: MGYG-HGUT-01384). The cgMLST scheme (cgMLST100) comprised 2600 core genes. Pairwise allelic distances were calculated using cgmlst-dists (v0.4.0) (https://github.com/tseemann/cgmlst-dists (accessed on 25 March 2024)). A minimum spanning tree (MST) was generated in GrapeTree (v1.5.0) [33] through the MSTreeV2 method.

### 2.6. Environmental Investigations

Guided by investigations on common exposures, lots of saline products that were delivered to at least two facilities with confirmed cases were analyzed. We requested information on production and quality control procedures from the manufacturers and searched for links to products from the manufacturer involved in the Australian outbreak. Retained samples were tested for sterility by the manufacturers.

We requested customer lists from the manufacturer involved in the Australian outbreak for lots delivered to Europe produced during the period of the Australian outbreak. We traced these lots by contacting respective national product regulatory agencies in the countries of supplied distributors and requesting customer lists of onward sales.

### 2.7. Ethics

The Robert Koch Institute (RKI) is a higher federal authority within the portfolio of the Federal Ministry of Health. According to §4 IfSG, the RKI as the National Public Health Institute has the task of epidemiologically analyzing the national data on notifiable infectious diseases and on notifiable detections of pathogens. This study was performed according to the aforementioned legal mandate of the RKI. In this context, no ethical review is necessary.

## 3. Results

### 3.1. Case Findings and Characterization

Overall, we identified 25 initially possible cases from August 2023 to June 2024 (Figure 1). Of these, 9 cases were identified via positive blood cultures documented in ARS (36%), 14 through the introduced national surveillance (56%), and 2 through both (8%).

Isolates were available for sequencing for 18/25 cases (72%, Figure 2). From these, 6/18 cases were confirmed because isolates clustered within a 6-allele difference. We excluded 11/18 cases because of substantially higher allele differences and 1/18 because sequencing confirmed the isolate to be *R. insidiosa*. Seven of the twenty-five cases (18%) remained probable cases as these were retrospectively identified through *R. pickettii*-positive blood cultures reported in ARS, but no isolates could be retrieved for sequencing.

Confirmed, probable, and excluded cases had similar age distributions, with an overall median age of 68 years (interquartile range: 60–73 years, Table 1). Men and women were similarly affected among confirmed and probable cases while men dominated among excluded cases.

The six confirmed cases occurred in three hospitals across three federal states between October 2023 and March 2024. In October 2023, three cases occurred in hospital 1 and one case occurred in hospital 2. One case in February 2024 occurred in hospital 1, and one case in March 2024 occurred in hospital 3. The confirmed cases from different hospitals were not linked by patient transfer. For all confirmed cases, indwelling devices were applied and exposure to saline solutions was documented. Apart from one case with surgical site infection, all confirmed cases showed signs of systemic infection, cardiac decompensation, tachycardia, or sepsis. One death was reported.

In June 2024, we declared the end of the outbreak as no further case had been confirmed in the last 3 months despite continuous surveillance and no probable cases were reported in ARS (as of 31 December 2024).

### 3.2. Microbiological Investigation

Genome sequences from isolates from the six confirmed cases in Germany differed within six alleles from the most similar sequence, and pairwise allelic differences spanned 0–9 alleles (Figure 2). The sequences from isolates of the other *R. pickettii* cases in Germany had ≥2457 allelic differences from the outbreak cluster. The minimal allele distance between the German cluster and the UK and Australian outbreak sequences was 43 alleles.

### 3.3. Environmental Investigation

Comparing prior exposures to medicines, procedures, indwelling devices, and medical products between confirmed cases, exposure to saline solutions, either as irrigation solution, preparation for medicines, or wound management was the only common link between all six confirmed cases in Germany. No medicines or procedures could be identified that were applied in all cases. Confirmed cases in hospitals 1, 2, and 3 were potentially exposed to solutions from manufacturers A and B. Only cases in hospital 1 were potentially exposed to solutions from manufacturer C. Four products by two manufacturers matched across the three hospitals, but lot numbers only matched in two of the three hospitals (Table 2).

Sterility tests by two manufacturers, A and B, on lot numbers 5 and 11, respectively, remained negative. Sterility tests on additional lots, i.e., another lot from manufacturer A and three lots from manufacturer B remained negative, too. All three manufacturers denied having been supplied with products from the manufacturer associated with the Australian outbreak.

The trace-forward of products from the manufacturer of the saline solution associated with the outbreak in Australia revealed that it usually distributes saline solution to a distributor within the European Union (EU) but outside of Germany. This downstream distributor stated that it had not sold any of the products within Germany.

## 4. Discussion

National surveillance sensitively detected an outbreak of *R. pickettii* in Germany from August 2023 to June 2024, upon an international alert in December 2023 on a concurrent outbreak in Australia. While clinical detection of *R. pickettii* is usually rare, particularly in blood, the ARS system in Germany had captured an increase in detections since August 2023. Six cases from three different hospitals and federal states belonged to one outbreak cluster Appendix A, confirming the outbreak in Germany.

Upon the international alert, ARS was able to swiftly detect a case accumulation confined in time and dispersed geographically, despite low case numbers. Legislation by IfSG law allowed the quick establishment of nationwide surveillance. In parallel, the infection control department in one hospital with several cases had already begun investigating a local cluster [36]. After communicating the German outbreak to EU states and the UK, the latter identified cases with isolate sequences matching the Australian outbreak clone and the same 0.9% sodium chloride product being available on the UK market [31]. The outbreak thus exemplifies how ARS, notification-based surveillance, local investigations, and international communication offer a powerful tool to sensitively detect and investigate outbreaks.

Many identified possible cases were subsequently excluded from the outbreak, mostly in material other than blood. This might reflect the raised awareness following the notification mandate and active outreach for isolate submission.

The only common link between the six confirmed cases in the German outbreak was exposure to saline solutions. This aligns with investigations within and between the hospitals with confirmed cases, as, for example, Krone et al. described for the local investigation [36]: affected patients were linked neither by place, personnel, or any medical device. The reported link based on electrolyte solution did not extend to the subsequently identified cases as part of this report. The confirmed case who was diagnosed in February 2024 in hospital 1 had a wound infection after surgery in October 2023 in the same hospital and time as three other cases. Thus, five of the six confirmed cases were exposed in October 2023. The concentration of cases/exposure in August-November 2023 would be in line with the hypothesis of temporary contamination of production lines. Despite high turnover, a contaminated product lot could still have led to the confirmed case exposed in March 2024.

For all confirmed cases, sodium chloride was applied either as medicine (e.g., intravenous infusion) or as a medical product (e.g., as an irrigation solution to place an intravenous line, to prepare medicines, or in wound care). An association of nosocomial outbreaks with saline solutions has been repeatedly reported [13,16]. Most hospitalized patients receive saline solutions, and there are only a few large manufacturers for saline solutions in Germany. For example, manufacturer A has a self-reported market share for saline solutions of 90%. Hence, we focused on identifying specific product lots applied to confirmed cases instead of association studies. While products from two manufacturers specified on the wards’ purchase lists matched between all three hospitals, no specific lots matched between more than two hospitals. We faced several challenges while searching for common product lots applied to all confirmed cases and forwarding them for sterility testing.

First, patient files usually do not specify the saline solutions, i.e., manufacturer and lot number. We consequently used delivery lists from the hospital pharmacies on the ward level (or hospital level, if lacking) around the time of exposure of the confirmed cases. This reduced specificity, and the workload caused further delay, particularly when these lists were available only in a paper-based form. Patient-level documentation of exposure to all medical products including lot numbers, at least if given intravenously, should be established to identify contaminated products/lots. This would necessarily require automated procedures, e.g., by scanning barcodes, to keep documentation efforts to a minimum. In the meantime, purchase and distribution lists should be available at least on the ward level, digitalised, and readily sharable and comparable between institutions.

Second, saline solutions have a high turnover in hospitals. The delay in identifying suspected lots means that suspected contaminated lots are not in stock at the time of investigation and cannot be tested for sterility. Sterility tests by the hospitals of several potentially affected lots or those in stock on affected wards at the time of investigation remained negative [36]. The investigation in the UK and several published *R. pickettii* outbreaks reported similar challenges in identifying contaminated lots [14,17,31].

Third, retesting of retained samples of product lots was challenging. Revisiting quality management processes with manufacturers revealed the common procedure of ‘parametric release’. Sterility testing is performed when the production line starts. Lots are subsequently not tested for sterility if indirect parameters, such as temperature, are within accepted ranges. Retained samples per lot are not guaranteed. Other outbreaks with *Ralstonia* spp. in 2023 associated with medicines and medical products indicate that gaps in the sterilization of these products are possible [37]. In the absence of an official mandate for sterility testing, the tests on convenient samples were performed by the manufacturers themselves, confirming sterility months after lot identification.

Fourth, saline solution, when used as an infusion or irrigation solution, has the same content but is classified as a medicine or medical product (or technically ‘medicinal product’ or ‘medical device’), respectively. The difference might not be obvious to clinical staff, but different regulations apply. Also, responsibilities vary within the regulatory authorities, which added to the communication and coordination efforts required in the outbreak investigation.

Among nosocomial *R. pickettii* outbreak investigations that identified contamination, many were due to medicines (above all, fentanyl) [18,19,20]. Other investigations could not identify any candidate source of infection [11,14] or hypothesized sources such as dialytic osmosis water or non-sterile gloves but could not report growth of *R. pickettii* from respective samples [10,15]. There are reports that traced outbreaks of *R. pickettii* bloodstream infections back to saline solutions and identified contaminated products. The reported outbreaks were rather large (11–30 cases) and occurred at single hospitals, which may have facilitated product identification [12,13,16]. In one of these outbreaks, the saline solution was an injection solution, and the use of a specific lot was documented for all cases [12]. Chen et al. describe immediate and extensive testing of items as key [13], and moreover, Bedir Demirdag et al. point out that from ten sampled saline bottles, growth was only successful in one [16]. Another investigation suspected saline solution as the source, but all tested samples remained negative [17].

Throughout the course of the outbreak in Australia, the country’s Therapeutic Goods Administration (TGA) issued a recall notice for all batches and additional products from the internationally distributing manufacturer [38,39,40,41]. Trace-forward of the saline solutions associated with the outbreaks in Australia and UK did not identify a link between affected lots and Germany. Nevertheless, distribution via sub-distributors cannot be ruled out. The contamination leading to the Australian and UK cases was most probably caused by a leaking safety seal in the production process of one manufacturer [42].

The cluster of cases in Germany differed from the simultaneously occurring cluster in Australia and UK by 43 alleles, whereas German outbreak cluster isolates differed between each other within 6 alleles. Nevertheless, the difference between the German and the Australian and UK outbreak cluster is orders of magnitude smaller than the difference between the German outbreak cluster and sequences from other patient and environmental samples in Germany (≥2457 allelic difference). On the other hand, the isolates from the UK and Australian investigations are genetically more similar to Dutch isolates (one from 2003 and two from 2012) than to the German concurrent outbreak. Hence, the current molecular data appears insufficient to confirm or dismiss a link between German, Australian, and UK cases. No proof could be established that the same manufacturer caused all three outbreaks, although the concurrent occurrence of the outbreaks might still be a strong signal pointing to a common source.

## 5. Limitations

We might underestimate the outbreak in Germany. In December 2023, we only included probable cases with positive blood cultures in the ARS system retrospectively, and ARS does not cover all clinical/microbiology laboratories or hospitals in Germany. Additionally, although widely shared, the alert in January 2024 that all detections of *R. pickettii* since September 2023 were notifiable might not have reached all laboratories and clinicians in Germany. Finally, pathogenicity (and testing upon symptoms) is largely limited to immunocompromised patients.

We confirmed the outbreak and cases by genetic clustering in the absence of an established cut-off and cgMLST scheme. The choice of the cluster of cases in Germany within six allelic differences is considered justifiable given the large distance from other isolates from Germany or the public domain included in the analysis.

The distances between sequences in this study and those from other studies might, at least partly, be affected by low sequencing quality and batch effects (e.g., differences caused due to sequencing platform, sample preparation, assembly, coverage, etc.). In the UK investigation [31], one isolate had low sequencing yield and SNPs were discounted manually, which might explain why Saunders et al. [31] report 0–4 SNP among their isolates, compared to the 1–30 allelic differences between them in our analysis. In the previous investigation with a smaller sample set by Krone et al., the difference between the Australian and German clusters was 26 alleles [36]. This highlights how genomic cluster information is dependent on the sample set, sequencing quality, and bioinformatic analysis methods. Hence, cluster information needs to be interpreted together with epidemiological findings.

Secondary genomic species determination identified multiple *Ralstonia* species among excluded cases, while they had been identified as *R. pickettii* by standard diagnostic methods (see Appendix A for further details). These inconsistencies reflect the limited knowledge about *Ralstonia* spp. that is subject to current research and development [43].

Exposure to specific saline solution products and lots could not be confirmed on the patient level, only on the ward or hospital level.

## 6. Conclusions

We found a supra-regional *R. pickettii* outbreak in Germany in 2023–2024. Although the outbreak vehicle could not be identified, saline solution, either as a medicine or as part of a medical product, probably caused the outbreak. Voluntary laboratory-based surveillance complements mandatory notification, especially for rare, non-notifiable organisms. Outbreak signals and suspected contamination of medical products should be communicated swiftly to responsible and public health authorities to enable effective prevention and control measures.

Challenges remain in the proof and control of outbreaks associated with contaminated products, particularly medical products (in contrast with medicines), as responsibilities differ within but also between the EU, national, and regional levels. Automated patient-level documentation of exposure to all medical product lots including saline solutions or digitalised purchase lists could facilitate investigations. However, storage and legal mandates for official testing of retain samples could substantially accelerate identification of contaminated products in future outbreaks.

## Figures and Tables

**Figure 1 microorganisms-13-02102-f001:**
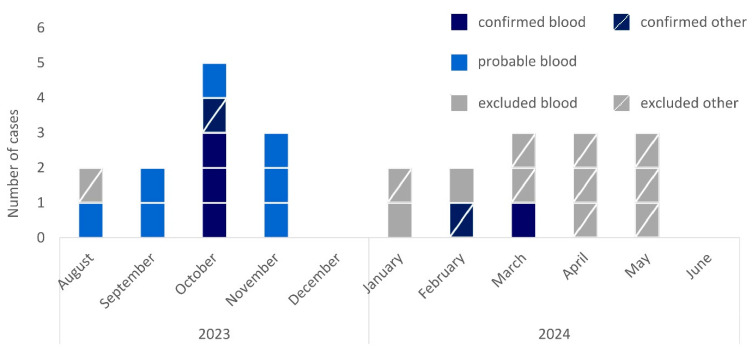
Epidemic curve of cases of *R. pickettii* in Germany by month of sampling, case category (confirmed, dark blue; probable, light blue; excluded, grey), and material (not crossed, blood culture; crossed, other material), August 2023–June 2024 (*n* = 25).

**Figure 2 microorganisms-13-02102-f002:**
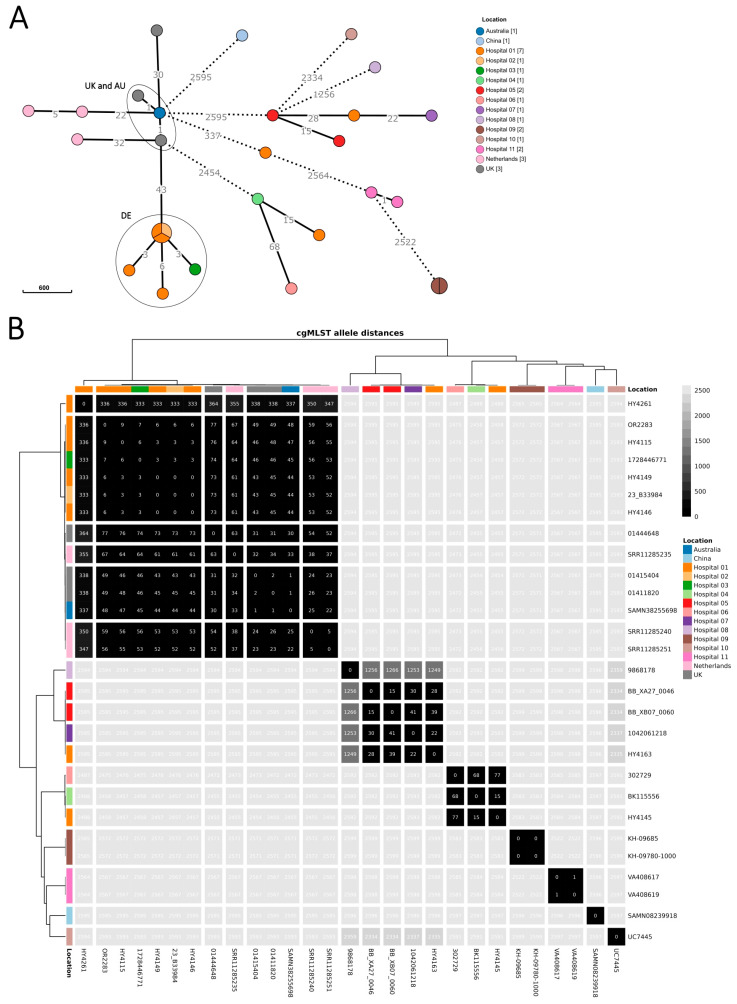
Minimum spanning tree of *R. pickettii* sequences (*n* = 28 isolates), upper panel (**A**), and heatmap showing pairwise allelic differences and dendrograms (hierarchical clustering with complete linkage based on Euclidean distances), lower panel (**B**). Isolates collected in Germany as part of the investigation between August 2023 and June 2024 (*n* = 20 isolates) are coloured by sampling location, i.e., the submitting hospital (or laboratory, if hospital was unknown). Included international sequences from the Australian outbreak and the UK investigation (‘UK and AU’ ellipse) and public repositories (*n* = 8 isolates) are coloured by country (for collection year, see Appendix A). The 20 isolates as part of the investigation comprise those of the confirmed and excluded cases (*n* = 18 isolates) and 2 samples from environmental screening. Locations 1–3 correspond to the 3 hospitals with the six confirmed cases of the German outbreak cluster (‘DE’ ellipse). Minimum spanning tree (panel **A**): branch lengths were on logarithmic scale and shortened/dashed for ≥100 allelic differences. Heatmap (panel **B**): identified genomic clusters are separated by white lines, visualized in R (v4.3.0) [34] with the *tidyheatmaps* package (v0.2.1) [35].

**Table 1 microorganisms-13-02102-t001:** Characteristics of *R. pickettii* cases by case definition in Germany, August 2023–June 2024 (*n* = 25). Proportions were calculated among the cases with respective information.

Patient Characteristics	Confirmed (*n* = 6)	Probable (*n* = 7)	Excluded (*n* = 12)	Total (*n* = 25)
Age in years				
Median (IQR ^a^)	68 (64–69)	69 (65–73)	67 (45–77)	68 (60–73)
Unknown—*n*	1	0	8	9
Sex—*n* (%)				
Female	2 (40%)	3 (43%)	1 (17%)	6 (33%)
Male	3 (60%)	4 (57%)	5 (83%)	12 (67%)
Unknown	1	0	6	7
Material—*n* (%)				
Autologous stem cells	1 (17%)	0	0	1 (4%)
Bronchoalveolar lavage	0	0	1 (8%)	1 (4%)
Blood	4 (67%)	7 (100%)	2 (17%)	13 (52%)
Cerebrospinal fluid	0	0	3 (25%)	3 (12%)
Joint/bone	0	0	2 (17%)	2 (8%)
Screening, not further specified	0	0	2 (17%)	2 (8%)
Urine	0	0	2 (17%)	2 (8%)
Wound	1 (17%)	0	0	1 (4%)

^a^ IQR, interquartile range.

**Table 2 microorganisms-13-02102-t002:** Potential common exposures to saline solutions of confirmed *R. pickettii* cases in Germany, August 2023–March 2024 (*n* = 6). ‘x’ marks possible exposure of cases in a hospital; ‘-’ indicates that products by the manufacturer were not purchased.

Manufacturer	Product	Lot	Hospital 1 (*n* Cases = 4)	Hospital 2 (*n* Cases = 1)	Hospital 3 (*n* Cases = 1)
Manufacturer A			x	x	x
	Product 1	Lot 1–3	x	-	x
	Product 1	Lot 4	-	x	x
	Product 1	Lot 5	x	x	-
	Product 2	Lot 6, 7	x	-	x
	Product 3	Lot 8–10	x	-	x
Manufacturer B			x	x	x
	Product 4		x	x	x
	Product 4	Lot 11	x	x	-
Manufacturer C			x	-	-

## Data Availability

All sequence data collected throughout the outbreak are available through the European Nucleotide Archive ENA (PRJEB96513).

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
