# Peer review of "Nosocomial Outbreak of Ralstonia pickettii Infections Likely Linked to Saline Solutions in Germany from August 2023 to March 2024—Challenges in Medical Product-Related Outbreaks"

_microorganisms, 2025, doi:10.3390/microorganisms13092102_

Round 1

Reviewer 1 Report

Comments and Suggestions for Authors

The manuscript is very well-written and organized. The work is well-performed and well-presented. It investigates the possibility of identifying nosocomial outbreak of Ralstonia pickettii infections in Germany following its reports in other countries. The study reports an important topic of rising concern in health care facilities which worth publishing. I have the following comment:

Regarding the definitions, please clarify that the definition refers to a “confirmed outbreak case” rather than only a “confirmed case” in line 79.

In the methods section, please clarify how was the microbiologic identification of suspected or probable cases performed. How was Ralstonia pickettii species been identified in probable cases before sequencing?

Line 106: Please correctly format the subtitle “Microbiologic investigation”

Lione 318: Please refer to the mentioned year of isolation (2003-2012) relationship in the figure as the temporal relation is not clarified.

Supplementary Figure 2 may be better transferred to the main manuscript as Figure 2 as it represents an important figure.

Reviewer 2 Report

Comments and Suggestions for Authors

The manuscript addresses a highly important issue of healthcare-associated infections caused by environmental, non-fermenting organisms with an often unclear source and route of transmission. The paper is carefully prepared, but several aspects require clarification or further explanation.

The case classification raises some questions:  why were only blood isolates unavailable for sequencing considered probable cases? What was the rationale behind this categorization?

Table 1 suggests that only one isolate was obtained per case. Was Ralstonia pickettii really detected in only one sample per patient? Were there no cases with multiple isolates from different clinical materials, and if so, were these genetically identical? What does “screening material” refer to in this context? It would be useful to include more clinical data in the table,  at the very least, information on patient survival or death.

To improve clarity, Figure 1 (the epidemic curve) should include the hospital from which each case originated. Among the six confirmed cases, four occurred in one hospital and one each in two other hospitals. There is a several-month gap between the first four cases in October and the next two in February and March. However, the authors state: 

“The confirmed case who was diagnosed in February 2024 had presumably been exposed in October 2023 in the hospital with three cases in that month. Thus, five of the six confirmed cases were exposed in October 2023.”

This sentence is unclear. Did the first four cases all occur in the same hospital? Was potential patient transfer between hospitals analyzed?

The “Environmental investigation” section and Table 2 also raise questions. The text states:

“Cases in hospitals 1, 2, and 3 were exposed to solutions from manufacturers A and B,”

whereas the table caption reads:

“‘x’ marks possible exposure of cases in a hospital.”

It should be clearly defined whether exposure was confirmed at the individual patient level or merely based on the presence of the product in a ward or hospital.

If exposure was not confirmed at the patient level, and given that contamination of any product was not proven, one should be very cautious when drawing conclusions about the source of infection. The suggestion in the title   “Ralstonia pickettii infections likely linked to saline solution”  may be too strong.

The discussion would also benefit from reference to other outbreaks caused by environmental organisms (e.g., other Ralstonia outbreaks) and the challenges they pose.
